# Aspirin Inhibits the Inflammatory Response of Protease-Activated Receptor 1 in Pregnancy Neutrophils: Implications for Treating Women with Preeclampsia

**DOI:** 10.3390/ijms232113218

**Published:** 2022-10-30

**Authors:** Scott W. Walsh, Marwah Al Dulaimi, Jerome F. Strauss

**Affiliations:** 1Department of Obstetrics and Gynecology, School of Medicine, Virginia Commonwealth University, Richmond, VA 23298-0034, USA; 2Department of Physiology and Biophysics, School of Medicine, Virginia Commonwealth University, Richmond, VA 23298-0034, USA

**Keywords:** aspirin, preeclampsia, pregnancy, neutrophils, protease-activated receptor 1, nuclear factor-kappa B, cyclooxygenase-2, thromboxane

## Abstract

Neutrophils expressing cyclooxygenase-2 (COX-2) extensively infiltrate maternal blood vessels in preeclampsia, associated with vascular inflammation. Because pregnancy neutrophils also express protease-activated receptor 1 (PAR-1, F2R thrombin receptor), which they do not in non-pregnant subjects, they can be activated by proteases. We tested the hypothesis that aspirin at a dose sufficient to inhibit COX-2 would reduce inflammatory responses in preeclampsia neutrophils. Neutrophils were isolated from normal pregnant and preeclamptic women at approximately 30 weeks’ gestation. Normal pregnancy neutrophils were treated with elastase, a protease elevated in preeclampsia, or elastase plus aspirin to inhibit COX-2, or elastase plus pinane thromboxane, a biologically active structural analog of thromboxane and a thromboxane synthase inhibitor. Preeclamptic pregnancy neutrophils were treated with the same doses of aspirin or pinane thromboxane. Confocal microscopy with immunofluorescence staining was used to determine the cellular localization of the p65 subunit of nuclear factor-kappa B (NF-κB) and media concentrations of thromboxane were measured to evaluate the inflammatory response. In untreated neutrophils of normal pregnant women, p65 was localized to the cytosol. Upon stimulation with elastase, p65 translocated from the cytosol to the nucleus coincident with increased thromboxane production. When neutrophils were co-treated with aspirin or pinane thromboxane, elastase was not able to cause nuclear translocation of p65 or increase thromboxane. In untreated neutrophils of preeclamptic women, the p65 subunit was present in the nucleus and thromboxane production was elevated, but when preeclamptic neutrophils were treated with aspirin or pinane thromboxane, p65 was cleared from the nucleus and returned to the cytosol along with decreased thromboxane production. These findings suggest that COX-2 is a downstream mediator of PAR-1 and demonstrate that PAR-1- mediated inflammation can be inhibited by aspirin. Given the extensive and ubiquitous expression of PAR-1 and COX-2 in preeclamptic women, consideration should be given to treating women with preeclampsia using a dose of aspirin sufficient to inhibit COX-2.

## 1. Introduction

Low-dose aspirin (81 mg/day), which primarily inhibits cyclooxygenase-1 (COX-1), is now standard of care to prevent preeclampsia in women who have risk factors [1]. However, consideration should be given to using a higher dose of aspirin that inhibits cyclooxygenase-2 (COX-2) to treat women with preeclampsia. Over 40 years ago, Goodlin et al. successfully treated a woman with hemolysis, elevated liver enzymes, and low platelet count (HELLP syndrome) with a dose of aspirin that inhibits COX-2 (600 mg three times a day) [2].

In recent years, many beneficial effects of aspirin have been reported that address pathophysiologic manifestations of preeclampsia [1,3]. For example, not only does aspirin inhibit platelet thromboxane (TX) production, but it also inhibits placental thromboxane production restoring the balance between thromboxane and prostacyclin [4]. Aspirin inhibits lipid peroxide induced by COX-2 derived superoxide anion, TNFα and thromboxane generated by COX-2 in pregnancy neutrophils [5,6,7]. It also decreases lipid peroxidation in the maternal circulation and the placenta [8,9,10,11].

Recent evidence suggests neutrophils play an important role in maternal vascular pathology in preeclampsia where they extensively and exclusively infiltrate maternal blood vessels (Figure A1) causing a sterile inflammatory response [3,12,13,14,15,16], which is much different from what occurs in the setting of infection [17]. This unusual response may relate to the fact that pregnancy neutrophils are different than neutrophils of non-pregnant subjects. In pregnancy, neutrophils express protease-activated receptor 1 (PAR-1, also known as F2R thrombin receptor), which they do not in non-pregnant subjects [18,19]. This means they can respond to stimuli they otherwise would not, such as proteases which are elevated in the maternal circulation in preeclamptic women [20,21,22,23,24,25,26]. Proteases that are elevated in preeclampsia and activate PAR-1 include neutrophil elastase, matrix metalloproteinase-1 and thrombin [20,21,22,23,24,25,26]. The extensive maternal vascular infiltration of neutrophils in preeclampsia is evident by their increased expression of PAR-1 (Figure A1).

Nuclear factor-kappa B (NF-κB) is a key driver of inflammation [27]. In the inactive state, it is localized in the cytosol, but upon phosphorylation by RhoA kinase, the p65 and p50 subunits of NF-κB move from the cytosol into the nucleus where they bind to transcription factor binding sites to increase expression of inflammatory genes [27]. We previously showed that protease activation of PAR-1 in neutrophils of normal pregnant women causes nuclear localization of the p65 subunit, mimicking the nuclear localization present in untreated neutrophils of preeclamptic women [20].

Given the unique characteristics of pregnancy neutrophils relating to protease activation and inflammation, and the beneficial effects reported for aspirin administration, we tested the following hypotheses: (1) a dose of aspirin that inhibits COX-2 will reverse protease induced activation of neutrophils obtained from normal pregnant women, and (2) the same dose of aspirin will reverse the activation present in neutrophils of preeclamptic women.

## 2. Results

Figure 1 shows representative confocal microscopy images of the localization of the p65 subunit of NF-κB (red) in pregnancy neutrophils treated with elastase alone and in the presence of aspirin to inhibit COX-2 or pinane TX to inhibit thromboxane synthase. In untreated control cells, p65 was localized to the cytosol, but upon stimulation with elastase, p65 translocated from the cytosol to the nucleus. When elastase was co-incubated with aspirin or pinane TX, p65 remained in the cytosol.

Figure 2 shows statistical analysis of the results for % nuclear localization of the p65 subunit of NF-κB in pregnancy neutrophils treated with elastase, elastase plus aspirin or elastase plus pinane TX. Elastase alone significantly increased the nuclear localization of p65, but when elastase was co-incubated with aspirin or pinane TX, p65 remained mostly in the cytosol, similar to control cells.

Figure 3 shows media concentrations of TXB2, a metabolite of TXA2, for pregnancy neutrophils treated with elastase, elastase plus aspirin or elastase plus pinane TX. Thromboxane production was consistent with the nuclear localization of p65. When elastase stimulated the translocation of p65 to the nucleus, TXB2 levels were significantly increased, but in the presence of either aspirin or pinane TX, elastase treatment did not significantly increase TXB2 levels with respect to untreated neutrophils.

Figure 4 shows representative confocal images of the localization of the p65 subunit of NF-κB for preeclamptic pregnancy neutrophils treated with aspirin or pinane TX. In untreated neutrophils, p65 was present in the nucleus indicating the neutrophils were activated. However, when preeclamptic neutrophils were incubated with aspirin or pinane TX, the amount of p65 in the nucleus was reduced and the amount of p65 in the cytosol was increased.

Figure 5 shows merged confocal immunofluorescent images for entire microscope lens fields containing multiple cells for preeclamptic pregnancy neutrophils. In untreated preeclamptic cells, the p65 subunit was present in the nucleus in almost every cell, whereas in aspirin treated cells, the amount of p65 in the nucleus was greatly reduced. These images demonstrate that the activation is extensive, as is the effect of aspirin to empty the nucleus.

Figure 6 shows statistical analysis of the results for % nuclear localization. Both aspirin and pinane TX significantly reduced the nuclear localization of p65 of untreated neutrophils.

Figure 7 shows the TXB2 media concentrations for neutrophils obtained from preeclamptic subjects. There was considerable variation among subjects, which appeared to be due to the use of aspirin. The lowest TXB2 concentration was for a subject who had mild preeclampsia. The next lowest value was from a subject on low-dose aspirin before and during her pregnancy for congenital heart disease. None of the subjects with the highest TXB2 levels were on low-dose aspirin. Despite these varied values, in each case, aspirin and pinane TX decreased TXB2 production, although the TXB2 concentration for pinane TX did not reach statistical significane.

## 3. Discussion

We obtained support for our first hypothesis that elastase, a protease that is elevated in the maternal circulation of women with preeclampsia, activated neutrophils of normal pregnant women, causing the translocation of the p65 subunit of NF-κB from the cytosol to the nucleus. Coincident with nuclear localization of p65, there was a significant increase in thromboxane, an NF-κB regulated gene product and a metabolite of cyclooxygenase activation. When elastase was co-incubated with aspirin to inhibit COX-2 or with pinane TX to inhibit thromboxane synthase, the p65 subunit of NF-κB remained mostly in the cytosol and thromboxane was not increased as compared to untreated neutrophils, indicating that COX-2 and thromboxane are down-stream mediators of protease activation of PAR-1 in pregnancy neutrophils.

We also obtained support for our second hypothesis that aspirin reverses the activation present in neutrophils obtained from preeclamptic women. The p65 subunit of NF-κB was already present in the nucleus of untreated neutrophils, but upon treatment with aspirin or pinane thromboxane, the nucleus emptied and p65 returned to the cytosol, coincident with decreased thromboxane production. The evidence of activation was present in almost all preeclamptic pregnancy neutrophils examined, and the evidence that aspirin reversed the activation was almost complete.

Circulating neutrophils in pregnant women express the inducible form of cyclooxygenase, COX-2, and its expression is significantly greater in preeclamptic women than in normal pregnant women [7]. Lipid peroxides are potent activators of neutrophils, so induction of COX-2 likely occurs as neutrophils circulate through the intervillous space and are exposed to lipid peroxides secreted by the placenta [28,29,30]. Aspirin inhibits placental production of thromboxane and lipid peroxides [10,11], so aspirin treatment would reduce neutrophil activation by the placenta, and presumably, their infiltration into maternal blood vessels. In women with preeclampsia, vascular infiltration of neutrophils is associated with increased expression of COX-2 in endothelial cells and vascular smooth muscle cells along with increased expression of thromboxane synthase and NF-κB [12,13,14,15]. Additionally, vascular expression of PAR-1 is increased in preeclampsia [31].

Beneficial effects of aspirin for prevention of preeclampsia are thought to relate to inhibition of platelet thromboxane production, but the effects must extend beyond platelets. Thromboxane is produced by many cell types, including neutrophils, other leukocytes, endothelial cells, vascular smooth muscle cells, and trophoblast cells of the placenta, all of which contribute to the pathology of preeclampsia, and all of which express COX-2 and PAR-1 [3].

RhoA kinase is a known mediator of the inflammatory response induced by PAR-1 [32,33], which is achieved by RhoA kinase phosphorylation of NF-κB causing the translocation of its subunits to the nucleus to increase expression of inflammatory genes [34,35]. To our knowledge, we have provided the first evidence that the PAR-1/RhoA kinase pathway involves cyclooxygenase and thromboxane as intermediates, and that protease activation of PAR-1 can be inhibited with aspirin.

Our study has clinical implications. PAR-1 is constitutively expressed in many cell types, including endothelial cells, smooth muscle cells, neurons, platelets, leukocytes, and trophoblast cells of the placenta [3,32,33,36,37,38] and PAR-1 activation can explain clinical and pathologic manifestations of preeclampsia as recently reviewed [1,3] (See Figure A2). For example, protease activation of PAR-1 can explain placental dysfunction related to oxidative stress, increased soluble fms-like tyrosine kinase 1 (sFlt-1) and increased thromboxane. Several studies show PAR-1 is expressed in the plaenta [36,37,38,39,40]. Placental oxidative stress can be explained by protease activation of PAR-1 expressed on trophoblast cells, which activates NADPH oxidase to generate reactive oxygen species [37]. PAR-1 activation can explain the elevation in angiogenic factors because trophoblast and decidual production of sFlt is stimulated by protease activation of trophoblast PAR-1 and NADPH oxidase [37,38,40]. Activation of NADPH oxidase can explain the imbalance of increased thromboxane and decreased prostacyclin which characterizes preeclampsia because oxidative stress drives the imbalance [41]. It can explain enhanced vascular reactivity to angiotensin II leading to hypertension [13]. It can also explain the neutrophil inflammatory response leading to vascular dysfunction [12,13,14,15]. Protease activation of PAR-1 on endothelial cells activates NF-κB, upregulates cell adhesion molecules, (ICAM-1), triggers production of neutrophil chemokines, (interleukin-8), and increases permeability of the endothelial cells by causing them to contract triggering edema [32,33,42,43,44]. PAR-1 contraction of renal endothelial cells would explain proteinuria. Finally, activation of PAR-1 on platelets leads to coagulation abnormalities [3,32,33].

Our finding that aspirin reverses the activation present in neutrophils of preeclamptic women is important for several reasons. First, it shows that treatment with aspirin may have an immediate effect to inhibit activation of circulating neutrophils to prevent their infiltration into the maternal vasculature. Second, it suggests aspirin treatment would inhibit activation of new neutrophils entering the circulation from the bone marrow. Third, since neutrophil elastase and other PAR-1 activating proteases are elevated in preeclampsia, initial neutrophil activation may result in a feed forward cycle where initial activation results in protease release which in turn activates other neutrophils. Such a scenario fits with the progressive worsening of clinical symptoms. Aspirin treatment would presumably break this vicious cycle. Forth, we previously showed that factors other than proteases, such as lipid peroxides, can activate pregnancy neutrophils and that aspirin and pinane TX inhibit lipid peroxide activation [5], so cyclooxygenase and thromboxane appear to be common mediators of neutrophil activation in pregnancy. Aspirin treatment would presumably also break this cycle.

Our study provides the first evidence that COX-2 and thromboxane are down-stream mediators of PAR-1, and that inhibition of COX-2 inhibits the ability of PAR-1 to activate NF-κB-mediated inflammation. These findings provide a rationale for treating women with preeclampsia using a higher dose of aspirin than what is used for prevention. A weakness of our study is that we did not demonstrate aspirin inhibits PAR-1 activation in cells other than pregnancy neutrophils. However, given that aspirin also inhibits PAR-1 activation by thrombin in platelets to prevent coagulation, cyclooxygenase is likely a common down-stream mediator of PAR-1 activation.

The potential benefit of aspirin for treating women with preeclampsia is suggested by the Goodlin study [2] which represents a classic case for the efficaciousness of a drug. This was a woman who had two previous preeclamptic pregnancies with poor outcomes and who developed HELLP syndrome in her third pregnancy. She was placed on aspirin and her symptoms improved, she was taken off aspirin and her symptoms worsened. She was placed back on aspirin and her symptoms improved. Goodlin et al. discontinued aspirin due to fear it would close the fetal ductus. This is why aspirin was declared to be contraindicated for use in pregnancy due to concern that it might cause closure of the ductus arteriosus or reduce amniotic fluid volume. However, these concerns may be unwarranted. Only 30% of an aspirin dose crosses from the maternal to the fetal side of the placenta [9], and, more importantly, the safety of aspirin was demonstrated by the Collaborative Perinatal Project in the 1970’s. Over 24,000 of the women in this study took aspirin during their pregnancy, 1500 of whom were heavily exposed. The Collaborative Perinatal Project found no harmful effects of aspirin use on the neonates [45]. Bleeding is a known risk factor for aspirin, so patients should be carefully monitored both antepartum and postpartum [46]. Treatment with aspirin would be best coupled with a calcium carbonate antacid to prevent stomach upset. Aspirin is inexpensive and would have the advantage of being readily available for treatment world-wide for preeclampsia.

Given (1) the widespread tissue and cellular expression of COX-2 and PAR-1 in preeclamptic pregnancy; (2) that COX-2 mediates the inflammatory effects of PAR-1; and (3) that activation of PAR-1 can explain clinical and pathologic manifestations of preeclampsia, consideration should be given to conducting a clinical trial to treat women with preeclampsia using a dose of aspirin that inhibits COX-2.

## 4. Materials and Methods

### 4.1. Study Subjects

Clinical characteristics of the study subjects are given in Table 1. Gestational age matched blood samples were collected from a multi-racial population at approximately 30 weeks’ gestation from women with normal pregnancy who went on to deliver at term (*n* = 11). In addition, blood samples were collected from one woman with mild preeclampsia who delivered at term and from four women with severe preeclampsia who delivered preterm (<37 weeks). Two 10 mL heparin tubes of blood were collected from each subject. Normal pregnant women had blood pressures ≤110/70 mmHg, no proteinuria, and no other complications. Preeclamptic women had blood pressures of ≥140/90 mmHg on 2 occasions at least 4 h apart after 20 weeks’ gestation and proteinuria (protein/creatinine ratio, mg/mmol, ≥0.3). Two of the preeclamptic women conceived by in vitro fertilization, and all had one or more risk factors, including chronic hypertension, pre-gestational diabetes, previous preeclampsia, gestational diabetes, and twin pregnancy. Only one of the women was on low-dose aspirin therapy. Vaginal deliveries were induced in preeclamptic women. None of the women were in labor at the time of sample collection. The Office of Research Subjects Protection of Virginia Commonwealth University approved this study (HM20009145). The procedures followed were in accordance with institutional guidelines and all subjects gave informed consent.

### 4.2. Neutrophil Cell Culture

Lymphocytes and monocytes were separated from granulocytes (96% of which are neutrophils) by Histopaque (1077/1119) density gradient centrifugation according to the manufacture’s protocol (Sigma-Aldrich, St. Louis, MO, USA) and as previously described [6,13,31]. For confocal microscopy studies, neutrophils were seeded at 200,000 cells per ml in Falcon 4-well cell culture slides (#354104) and cultured in Iscove’s Modified Dulbecco’s Medium supplemented with 10% fetal bovine serum and 1% antibiotics and antimycotics (100 U/mL penicillin, 100 µg/mL streptomycin, 0.25 µg/ mL amphotericin B) at 37 °C in a humidified 5% CO_2_ atmosphere. For measurement of media thromboxane B2 (TXB2), neutrophils were seeded at an average of 5,000,000 cells per mL. Neutrophils of normal pregnant women were incubated with the following treatments for 1 h for confocal microscopy studies and 2 h for media TXB2 measurement: (1) untreated control; (2) elastase (0.33 U/mL, Sigma-Aldrich); (3) elastase plus aspirin (ASA, 100 µM, acetylsalicylic acid, Sigma-Aldrich) to inhibit COX-2; and (4) elastase plus pinane thromboxane A2 (pinane TX, 10 µM, Cayman Chemical Company, Ann Arbor, MI, USA) to inhibit thromboxane synthase. We used elastase because it is a neutrophil product elevated in women with preeclampsia, and we previously showed that it activates pregnancy neutrophils via PAR-1 [20]. Aspirin is rapidly de-acetylated in the circulation, so we used a dose based on the plasma concentration of salicylic acid at 1.5 h after oral administration of 500 mg of aspirin [47], which approximates the dose used in the Goodlin study of 600 mg. Neutrophils of preeclamptic women were exposed to the following treatments: (1) untreated control; (2) ASA; and (3) pinane TX.

### 4.3. Confocal Microscopy Immunofluorescent Analysis

For confocal microscopy studies, media were collected, cells were washed with PBS and then fixed with 4% formalin for 1 h. Cells were washed, then blocked and permeabilized for 1 h in 10% goat serum + 3% BSA + 0.25% Triton X-100. Cells were incubated overnight at 4 °C with primary rabbit antibody specific for the p65 subunit of NF-κB (1:50, Proteintech, Rosemont, IL, USA). We evaluated the p65 subunit because it is localized to the nucleus in neutrophils of women with preeclampsia [20]. The next morning, cells were washed and incubated with anti-rabbit IgG secondary red fluorescence dye (Cy3, 1:2000, Jackson ImmunoResearch, Burlingame, CA, USA) at room temperature for 2 h. Cells were washed and mounted with VectaMount medium containing 4′, 6-diamidino-2-phenylindole (DAPI, Vector Laboratories, Burlingame, CA, USA) for nuclear DNA staining. Images were taken using a confocal microscope (Zeiss LSM 700, x63 lens, Dublin, CA USA). Depending on the number of neutrophils obtained from each subject, 2 to 7 replicates were performed for each treatment with an average of 7 cells per counting frame. The intensity of fluorescence in the nucleus vs. the cytosol was quantified using ImageJ software Version number 1.53 (imagej.nih.gov) using the Freehand tool and Integrated Density measurement. The density of the nucleus was taken as a percentage of the density for the whole cell.

### 4.4. Measurement of Media Thromboxane B2 (TXB2)

For measurement of TXB2 production, media were collected at the end of the 2 h incubation period and frozen at −20 °C until assayed. Neutrophil cell culture media were analyzed for TXB2 using a specific TXB2 ELISA kit (ADI-900-002) according to the manufacturer’s instructions (Enzo Life Sciences, Farmington, NY, USA).

### 4.5. Data Analysis

Subject demographic data are presented as means ± SD and experimental data are presented as box and whisker plots with error bars representing the 5 and 95% confidence limits. Data were analyzed by non-parametric Kruskal–Wallis test with Dunn’s multiple comparisons test using a statistical software program (Prism 9, GraphPad software, San Diego, CA, USA). The Friedman non-parametric test for matched data was used for the preeclamptic TXB2 results because the variation among patient samples was large. Treatment replicates were averaged for statistical analysis. Power analysis indicated a sufficient sample size to achieve 0.80 power to detect an α of 0.05. A *p* < 0.05 was considered statistically significant.

## 5. Conclusions

Our study presents new evidence that a dose of aspirin sufficient to inhibit COX-2 prevents the inflammatory response induced by protease activation of pregnancy neutrophils. To our knowledge, the finding that COX-2 is a downstream mediator of PAR-1 is novel and suggests that aspirin at a dose sufficient to inhibit COX-2 may be effective in treating women with preeclampsia. A summary of our findings is presented in Figure 8. Elevated levels of proteases in the maternal circulation of preeclamptic women activate neutrophils due to their pregnancy specific expression of PAR-1.

Activation of PAR-1 leads to increased expression of inflammatory genes. Aspirin by inhibiting COX-2, a downstream mediator of PAR-1, prevents the PAR-1 inflammatory response.

## Figures and Tables

**Figure 1 ijms-23-13218-f001:**
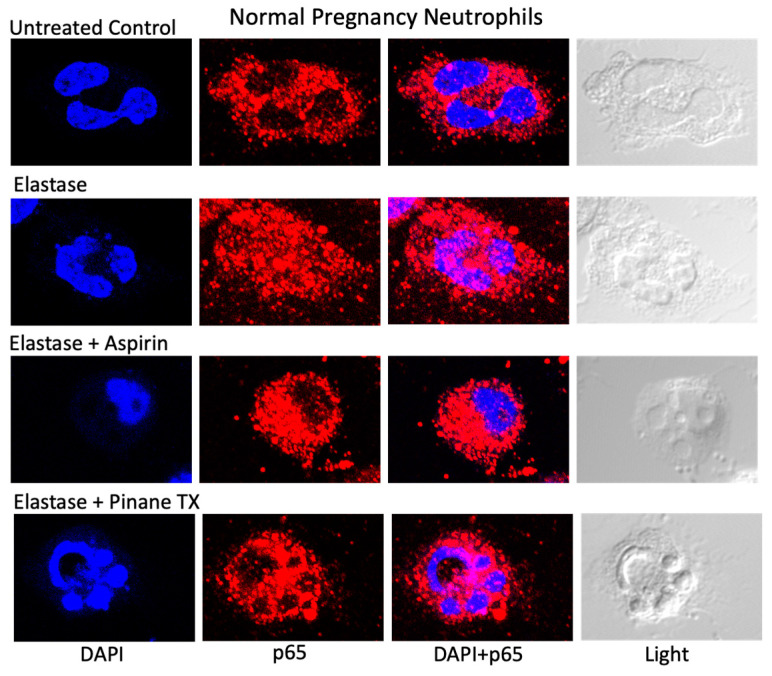
Confocal microscopy images. Red fluorescence, the p65 subunit of NF-κB; DAPI blue location of nucleus. x63 lens.

**Figure 2 ijms-23-13218-f002:**
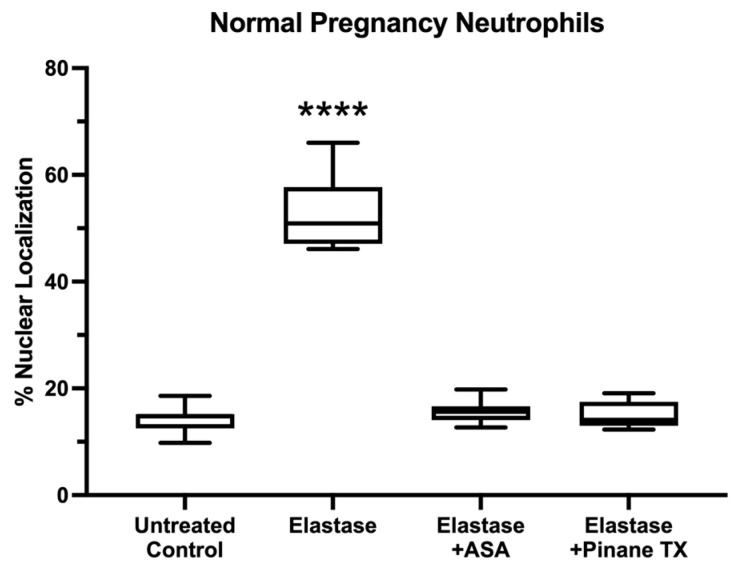
Percent nuclear localization of p65. (*n* = 11, **** *p* < 0.0001; ASA, aspirin).

**Figure 3 ijms-23-13218-f003:**
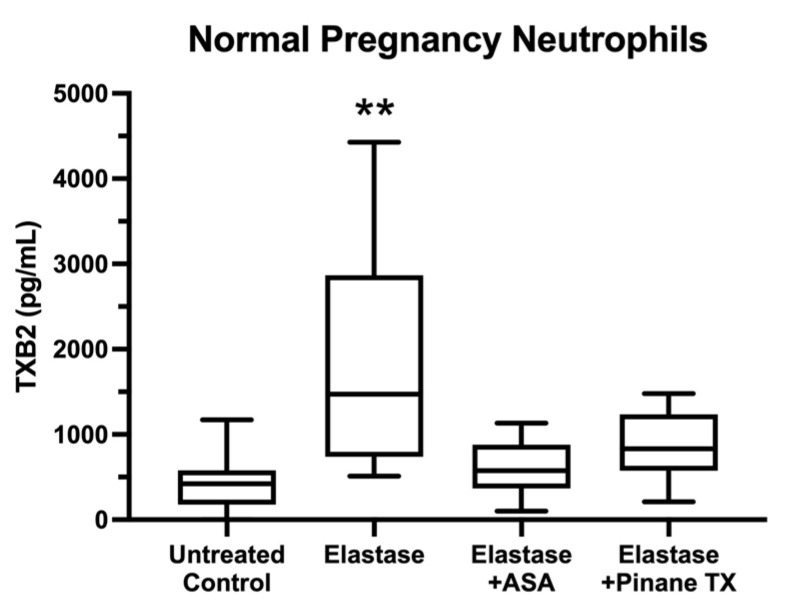
Media concentrations of TXB2. (*n* = 11, ** *p* < 0.01; ASA, aspirin).

**Figure 4 ijms-23-13218-f004:**
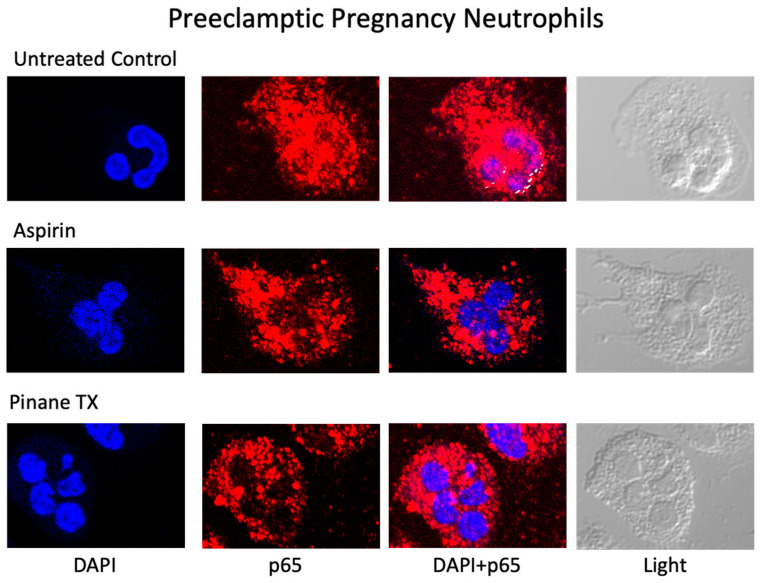
Confocal images of preeclamptic pregnancy neutrophils. Red fluorescence, the p65 subunit of NF-κB; DAPI blue location of nucleus. x63 lens.

**Figure 5 ijms-23-13218-f005:**
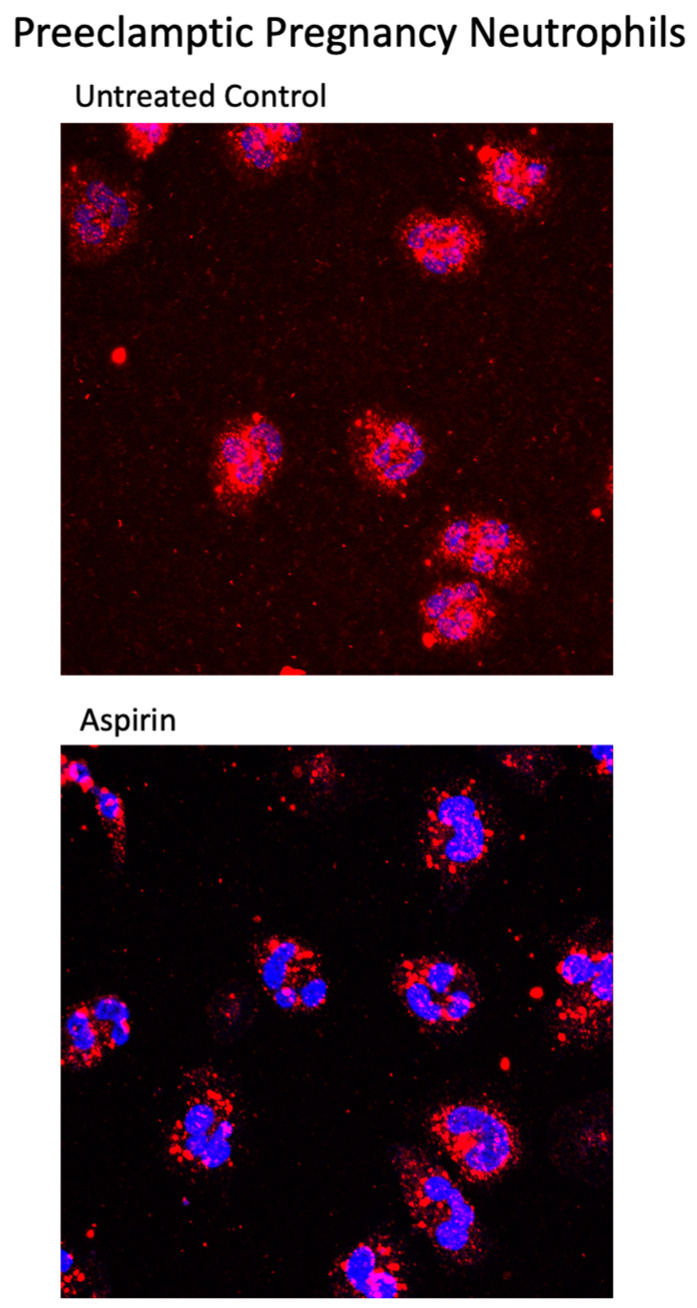
Confocal images for preeclamptic pregnancy neutrophils. Red fluorescence, the p65 subunit of NF-κB; DAPI blue location of nucleus. x63 lens.

**Figure 6 ijms-23-13218-f006:**
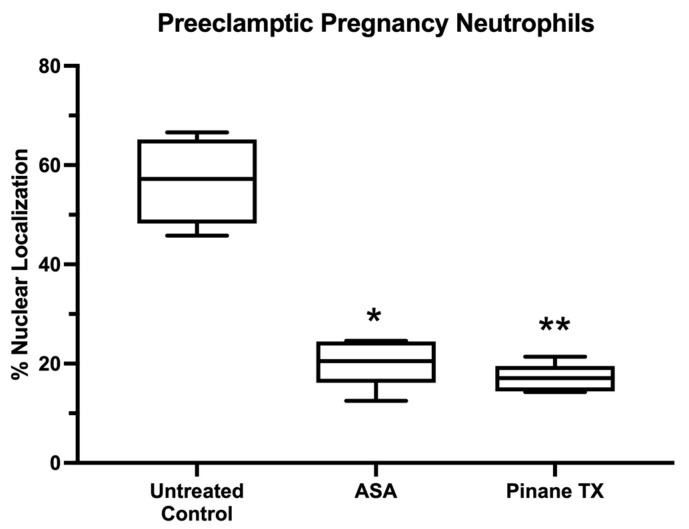
Percent nuclear localization of the p65 subunit of NF-κB in preeclamptic pregnancy neutrophils treated with aspirin or pinane TX. (*n* = 5, * *p* < 0.05; ** *p* < 0.01; ASA, aspirin).

**Figure 7 ijms-23-13218-f007:**
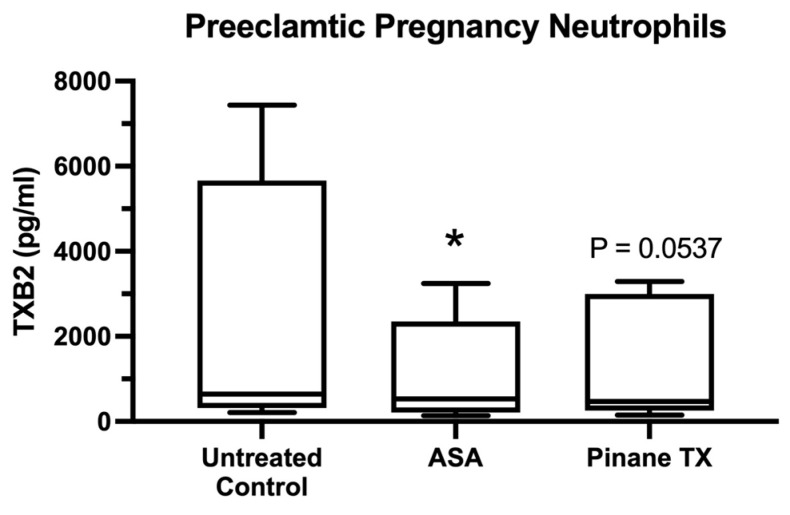
Media concentrations of TXB2 in preeclamptic pregnancy neutrophils treated with aspirin or pinane TX. (*n* = 5, * *p* = 0.0228; ASA, aspirin).

**Figure 8 ijms-23-13218-f008:**
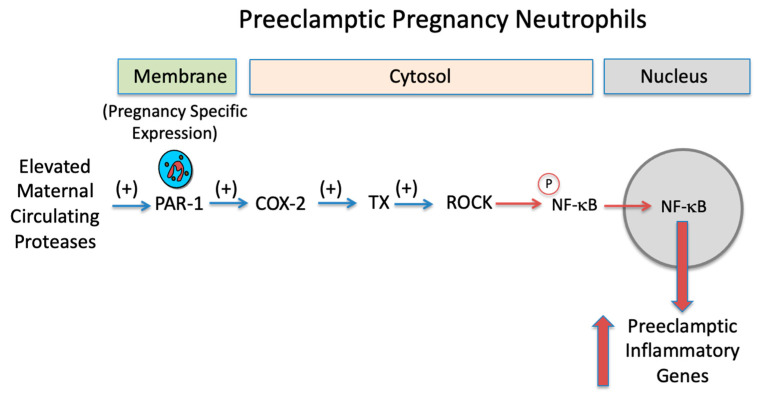
Mechanism for aspirin to inhibit neutrophil inflammatory response. Elevated levels of proteases in the maternal circulation of preeclamptic women activate the pregnancy specific expression of PAR-1 on neutrophils (**Top**). PAR-1 then activates COX-2 to increase thromboxane which activates RhoA kinase. RhoA kinase phosphorylates NF-κB which causes its translocation from the cytosol to the nucleus to increase the expression of inflammatory genes. Aspirin (**Bottom**), by inhibiting COX-2, prevents activation of the downstream mediators of PAR-1, so phosphorylation of NF-κB does not occur. NF-κB remains in the cytosol and does not translocate into the nucleus to increase preeclamptic inflammatory genes. Inhibition of PAR-1 downstream signaling is a new mechanism for the anti-inflammatory effects of aspirin. (TX, thromboxane; ROCK, RhoA kinase; P, phosphorylation; ASA, aspirin).

**Table 1 ijms-23-13218-t001:** Clinical Characteristics of Study Subjects.

Variable	Normal Pregnancy *n* = 11	Preeclamptic Pregnancy *n* = 5
Maternal age (years)	27 ± 3	33 ± 6
Pre-pregnancy BMI (kg/m^2^)	25 ± 5	31 ± 10
BMI at sample collection (kg/m^2^)	30 ± 6	37 ± 10
Systolic blood pressure at 30 weeks (mmHg)	109 ± 12	157 ± 11 ****
Diastolic blood pressure at 30 weeks (mmHg)	70 ± 10	92 ± 12 ***
Protein/creatinine Ratio	ND	0.56 ± 0.28
Primiparous	4	1
Multiparous	7	4
Race		
White	6	3
Black	5	2
Type of Delivery C-section Vaginal	1 10	3 2
Gestational age at sample collection (weeks)	30 ± 3	32 ± 4
Gestational age at delivery (weeks)	39 ± 1	33 ± 4 ****
Infant birth weight (grams)	3326 ± 371	1904 ± 672 ****

Values are mean ± SD. *** *p* < 0.001; **** *p* < 0.0001. ND = not determined.

## Data Availability

Data sharing is not applicable to this article.

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
