# Peer review of "Aspirin Inhibits the Inflammatory Response of Protease-Activated Receptor 1 in Pregnancy Neutrophils: Implications for Treating Women with Preeclampsia"

_ijms, 2022, doi:10.3390/ijms232113218_

Round 1
Reviewer 1 Report
The manuscript “Aspirin inhibits the inflammatory response of protease-activated receptor 1 in pregnancy neutrophils: implications for treating women with preeclampsia” presents in vitro evidence of the inhibitory effect of aspirin on the antiinflammatory effect of aspirin by reversing the elastase-induced neutrophil activation of preeclamptic women.
I found the methods used appropriate and well explained, as well as the experiment design.
Although the results are scientifically relevant, I suggest to revise the way you explained them in the Results section. Additionally, I suggest the following minor changes to the manuscript:
Introduction, line 20: It should be specified the role of pinane thromboxane, to make clear why you will include it in the experiment (it is a biologically active structural analog of thromboxane and a thromboxane synthase inhibitor).
Introduction, line 50: I think what you wanted to explain is that “aspirin inhibits lipid peroxidation induced by COX-2 derived superoxide anion”.
Introduction, line 50 bis: The abbreviation for thromboxane should be added.
Introduction, line 76: I suggest “ aspirin will reverse the elastaase-induced activation present…”
Results, line 81: If you consider that the abstract is not the right place to clarify the role of pinane thromboxane, you could do it in this first sentence of the Results section.
Results, lines 88 and 118: Figures 2 and 6 are not a summary of the results of confocal microscopy, but que quantification or the quantitative analysis of these results. Please exclude the word “summary” and write these sentences again.
Results, line 91: I suggest “ p65 remained “mostly” in the cytosol, as not the whole of p65 molecules remained in the cytosol.
Results, line 95: Please explain here that TXB2 is a metabolite of TXA2, in order to understand these results better.
Results, lines 98-99: I suggest “elastase treatment did not significantly increase TXB2 levels respect to untreated neutrophils”.
Results, lines 102-106: I think that this whole paragraph needs to be written in a different way. Terms such as “the nucleus emptied” and “p65 returned to the cytosol”, as you can not be sure whether the molecules were redirected to the cytosol or less molecules were translocated and returned. These are images and therefore show not a dynamic but a state event. Finally, in line 104 I suggest “p65 was localized mostly in the nucleus” as not all the molecules but a percentage of them were localized there.
Results, lines 110-114: This paragraph needs to be written in a different way too, avoiding terms such as “in every cell”, “was empty”, “to empty the nucleus”.
Results, lines 123-127: please change “TXB2 results” and “TXB2 values” for “TXB2 levels/concentration levels”.
Results, line 128: the decrease in TXB2 in the Pinane TX experimental group is not statistically significant.
Discussion, lines 143-149: Please write this paragraph again, taking into account the comments made above on the Results section.
Conclusions, lines 305-308: I suggest writing this paragraph again. The first sentence is in fact a conclusion from the results presented in this work but not the second one. It is previous knowledge. You can keep the information but make emphasis on the novelty of your results or what previous knowledge they are in line with.
Reviewer 2 Report
The article "Aspirin Inhibits the Inflammatory Response of Protease-Activated Receptor 1 in Pregnancy Neutrophils: Implications for Treating Women with Preeclampsia" is devoted to the important problem of finding pathogenetic mechanisms in preeclampsia, as well as the search for drug therapy for this syndrome. The article was written using a number of molecular research methods.
Although the idea of ​​the article is original, the presentation of the text and presentation of the data needs to be improved. Thus, in the introduction and summary, the main focus is on the PAR-1 and COX2 receptors. Basically all the data. However, in the results, the data are given by the analysis of neutrophils and methods of their activation. There is a feeling of fragmentation of the text and some processing is required. Interestingly, the authors do not provide data on the state of the placenta, but provide an immunohistochemical study of omental fat vessels. In addition, when describing the methods, I did not find a study of biopsy specimens of the omentum. And when they were taken.
That seems not entirely logical and requires explanation and additions. The main remark is the lack of integrity of the article, the text is presented in some fragments (like a patchwork quilt)
Many literature references over 10 years old. Perhaps the authors should leave the key references, and replace the rest with newer ones.
Although the topic of the article is very relevant, I assume that it needs to be improved. I didn't quite time it.
It should also be noted that there was a large percentage of self-citations (more than 23 out of 41 sources).
Reviewer 3 Report
The study shows interesting and reliable results suggesting the potential effect of aspirin on COX-2.
The authors should review and correct the punctuation throughout the manuscript where they have included bibliographic references.
Abstract.
Line 16-17. "Neutrophils 16 were isolated from normal and preeclamptic pregnant women at 30 weeks' gestation".
Because in Table 1. Clinical Characteristics of study subjects the women included in this study were not all 30 weeks' gestation. It is suggested that the authors correct this statement at approximately 30 weeks' of gestation.
Materials and Methods.
Study Subjects.
Line 238-240. “Preeclamptic women had blood pressures of ≥140/90 mmHg on 2 occasions at least 4 hours apart after 20 weeks’ gestation and proteinuria (protein/creatinine ratio ≥0.3).”
Authors should include the units of measurement of the protein/creatinine ratio.
Authors should review the entire materials and methods section and describe consistently and homogeneously the reagents and consumables used in the study.
In table 1, could the authors include the hemoglobin values in both study groups?
Line 285-287. “The intensity of fluorescence in the nucleus vs. the cytosol was quantified using Image J software (NHI) using the Freehand tool and Integrated Density measure”
Authors must include the bibliographic reference of the software used.
Discussion.
Line 181-182. “Activation of PAR-1 on endothelial cells causes their contraction, which would explain edema and proteinuria. Finally, activation of PAR-1 on platelets leads to coagulation abnormalities”.
Authors should accompany this paragraph with a supporting bibliography.
Figure 8. Effect of Aspirin to inhibit Inflammation.
The authors are suggested to change the green arrow in the image to an arrow indicating COX2 inhibition by aspirin.
Reviewer 4 Report
This paper gives insights into the mechanisms by which aspirin acts in the context of preeclampsia, through an impact of neutrophils. The authors show that elastase treatment localizes the p65 NFKB protein into the nucleus, while aspirin prevents this (as well as Pinane TX, inhibitor of thromboxane A2), elastase being a protease that activates PAR-1 (F2R thrombin receptor), please use in the text the official symbol to prevent confusion. The concentration of TXB2 follows the same type of profile. In preeclampsia neutrophils, aspirin prevented NFKB p65 localization to the nucleus.
The results are innovative and straightforward.
Round 2
Reviewer 2 Report
Dear Authors!
I assume that the reviewer's comments are not taken into account, especially with regard to self-citation. 2-3 refer-ences to your own work is enough. There are more than 10 of them.
It should also be noted that one of the neutrophil activators is Phorbol-12-myristate-13-acetate Why didn't you use it, if it is traditionally the standard for neutrophil activation.